# A machine learning prediction model for waiting time to kidney transplant

Juliana Feiman Sapiertein Silva[1☯¤a], Gustavo Fernandes Ferreira[2☯¤b], Marcelo Perosa[3☯¤c], Hong Si Nga[1☯¤a], Luis Gustavo Modelli de Andrade[1☯¤a]*

1 Department of Internal Medicine—UNESP, Univ Estadual Paulista, Botucatu, Brazil, 2 Transplant Unit– Santa Casa Juiz de Fora, Juiz de Fora, Brazil, 3 Kidney-Pancreas Transplantation Service of Leforte and Oswaldo Cruz Hospitals, São Paulo, Brazil

☯ These authors contributed equally to this work.
¤a Current address: Transplant Unit Division, Botucatu, SP, Brazil
¤b Current address: Transplant Unit Division, Juiz de Fora, MG, Brazil
¤c Current address: Transplant Unit Division, Liberdade, São Paulo, SP, Brazil
* Gustavo.modeli@unesp.br

**Data Availability Statement:** The data are held in a public repository: https://www.kaggle.com/gustavomodelli/waitlist-kidney-brazil.

**Funding:** The authors received no specific funding for this work.

## Abstract

### Background

Predicting waiting time for a deceased donor kidney transplant can help patients and clinicians to discuss management and contribute to a more efficient use of resources. This study aimed at developing a predictor model to estimate time on a kidney transplant waiting list using a machine learning approach.

### Methods

A retrospective cohort study including data of patients registered, between January 1, 2000 and December 31, 2017, in the waiting list of São Paulo State Organ Allocation System (SP-OAS) /Brazil. Data were randomly divided into two groups: 75% for training and 25% for testing. A Cox regression model was fitted with deceased donor transplant as the outcome. Sensitivity analyses were performed using different Cox models. Cox hazard ratios were used to develop the risk-prediction equations.

### Results

Of 54,055 records retrieved, 48,153 registries were included in the final analysis. During the study period, approximately 1/3 of the patients were transplanted with a deceased donor. The major characteristics associated with changes in the likelihood of transplantation were age, subregion, cPRA, and frequency of HLA-DR, -B and -A. The model developed was able to predict waiting time with good agreement in internal validation (c-index = 0.70).

### Conclusion

The kidney transplant waiting time calculator developed shows good predictive performance and provides information that may be valuable in assisting candidates and their providers.

**Competing interests:** The authors have declared that no competing interests exist.

Moreover, it can significantly improve the use of economic resources and the management of patient care before transplant.

## Introduction

Brazil ranks first in number of kidney transplants in the world performed by Health Public System [1]. By July 2020, over 30,000 Brazilians were on waiting lists for a kidney transplant [2]. Although approximately 5,000 deceased donor kidney transplants take place yearly in the country, the supply of organs does not meet demand, and the gap is growing [2, 3]. As a result, recurrent tests and procedures are necessary every 2–3 years to maintain patients on active transplant list [4]. Since this poses a significant economic burden to the healthcare system, predicting a patient's waiting time can help planning for pre-transplant evaluation, and thus promote a more efficient use of resources in countries such as Brazil [5]. For example, full pretransplant evaluation would be carried out at registration in candidates with high chances of being transplanted, while those less likely to be transplanted on the short term would undergo only the most necessary tests at registration and have their evaluation completed 6–8 hours before surgery. Moreover, estimating waiting time on the transplant list can help identifying the underprivileged, and thus impact allocation score, bringing more equity to transplantation programs [6].

The time deceased donor recipients spend on a waiting list, has been typically expressed as the median waiting time [7]. However, it does not convey the risks of death or removal from the waiting list.

Hart et al. [8] have used a competing risk model to develop a waitlist outcome calculator that demonstrates the probability of outcomes on the waiting list, including waiting time. However, their model reportedly requires updating and the online version of their calculator is still not available online.

The machine learning approach can contribute to the development of robust predictive models [5]. It includes both conventional statistical analyses, as well as linear and logistic regression and non-linear models, such as decision trees, neural network, nearest neighbors, and support vector machine, which can capture non-linear relationships. Machine Learning uses an approach based on steps: obtaining the data, excluding unusual data ("outliers"), selecting variables, model train, and validation. These approaches allow fitting algorithms capable of making predictions [9–11]. Thus, this study aimed at identifying the relevant predictors and combine them into a predictor model to estimate time on a kidney transplant waiting list using machine learning.

## Methods

### Study design and population

This retrospective cohort study included data of patients registered, between January 1, 2000 and December 31, 2017, in the waiting list of São Paulo State Organ Allocation System (SP-OAS) /Brazil. SP-OAS has a database that holds over 10 years of information and provides a good sample of the Brazilian transplant population. Indeed, of 5,923 kidney transplants performed in the country in 2018, one third of them (2,095) were carried out in the state.

SP-OAS adopts a policy of regional allocation, centralized and controlled organ distribution, and decentralized organ procurement and harvesting. SP-OAS serves as the state´s organ transplantation system and operates a single database of the entire transplant population in the

state, but patients from the various transplant groups and dialysis centers are divided into 4 sub-regional waiting lists (FUNDERP; UNICAMP; UNIFESP; and HCFMUSP) according to location [12].

This study was approved by the Research Ethics Committee of the School of Medicine of Botucatu—UNESP (# 3.094.616; CAAE: 03660718.2.0000.5411). Informed consent was not required because the data were analyzed anonymously.

## Eligibility criteria

All patients registered during the study period in the SP-OAS waiting list for a kidney-alone transplantation were included.

Living-donor kidney transplant recipients and prioritized patients were excluded. Patients with missing data were also excluded (n = 51 missing subregional information).

## Allocation criteria

Allocation was performed as established by the National Transplantation System of the Brazilian Ministry of Health [12, 13]. For deceased donor transplants, allocation criteria are based on HLA matching (highest number of points for HLA DR, followed by HLA B and HLA A), recipient's age (<18 years), date of registration on the waiting list, and panel reactive antibody (PRA). A point score system based on blood group and HLA match is used as follows:

DR: 0 MM = 10 points; 1 MM = 5 points; 2 MM = 0 point;

B: 0 MM = 4 points; 1 MM = 2 points; 2 MM = 0 point;

A: 0 MM = 1 point; 1 MM = 0.5 point; 2 MM = 0 point.

Waiting time, allosensitization (cPRA >50), diabetes mellitus, and age < 18 years served as tiebreakers.

## Predictors

The following variables were evaluated as predictors: age, sex, race, comorbidities, time on dialysis, blood group, calculated panel class I (cPRA), HLA-A, HLA-B, HLA-DR, number of blood transfusions, pregnancies, previous kidney transplants, and pre-transplant serology for Hepatitis B and C.

**HLA frequency.** HLA frequency variables (A, B and DR) were calculated by dividing the number of times the allele of interest is observed in a population by the total number of copies of all the alleles at that genetic locus in the population. HLA frequency in the population of São Paulo state was obtained from the Allele Frequencies net database [14]. To standardize HLA antigen assignments an HLA Dictionary was used [15]. The serological equivalents were listed as expert assigned types.

## Outcomes

Deceased donor transplantation was considered as the primary outcome.

Secondary outcomes include death on transplant list, and removal from the waiting list. Reasons for list removal include withdrawn from treatment, kidney function recovery, refusal to undergo transplantation, additional listings for the same person, deteriorating clinical conditions, and transplant performed outside the state.

## Statistical analysis

Continuous variables were categorized according to a frequency histogram including- age and cPRA. Not normally distributed variables were transformed; the natural logarithm was applied to time on dialysis. HLA A, B, and DR were analyzed as continuous variables in order to reduce the number of categories. Homozygosis in HLA A, B, and DR were analyzed as independent variables. Variables with near-zero variance were removed from the analysis (Hepatitis C and Chagas's disease) as they are uninformative predictors and cause algorithm convergence problems at the modeling stage.

**Univariate analysis.** Univariate analyses were performed using the chi-square test for categorical variables and the Kurskall-Wallis test for continuous variables, considering transplant as the outcome. The analysis of the predictors associated with transplantation was performed using a Kaplan-Meier model (package survival, R software).

**Prediction model.** The dataset was randomly split into two subsets–derivation (training 75%) and internal validation (testing, 25%). A Cox regression model (package survival, software R) was fit with transplantation as the outcome. All predictors were included in the univariate model, and collinear factors were removed. Collinearity was measured using the variance inflation factor (VIF). VIF values > 3 indicated collinearity. Multivariate stepwise Cox regression was performed using the MASS package, with AIC ("Akaike information criterion") optimization. The lowest AIC indicated the final fitting model, which was used to develop the risk-prediction equations. Plots of scaled Schoenfeld residuals were used to check the proportionality assumption. Approximate proportional hazard effects were found in all cases. The final model was tested using 25% of the data. C statistics were used to assess the accuracy of prediction [16]. In addition, calibration plots were used to evaluate calibration for each model. Confidence intervals were reported according to the method of Louis and Zeger [17]. Sensitivity analyses were performed with Cox models fitted for each subregional list in order to address local variations. Regression analysis was also performed using competitive risk as described by Fine & Gray [18]. This analysis considers removal from the waiting list as a concurrent event (package cmprsk) [19]. Data analyses were performed using software R version 3.4.2.

## Results

The total number of cases retrieved was 54,055. After removing all records on living-donor transplants and prioritized patients, 48,153 registries were included in the final analysis. During the study period, 28.4% of the patients were transplanted with a deceased donor (**Fig 1**). The median waiting time for transplantation was 26.3 months.

Patients who underwent transplantation showed lower median age, shorter time on dialysis, and lower cPRA, as well as higher frequencies of HLA-DR, -B and -A. Transplant frequency was higher in blood groups A and AB, in patients with homozygosity in HLA-A, -B or -DR, and in the FUNDERP and UNICAMP subregions (**S1 Table**).

## Kaplan-Meier models

Kaplan-Meier univariate models show that the likelihood of transplantation was higher: in blood groups A and AB compared to groups B and O (**Fig 2A**); with zero cPRA compared to other cPRA values (**Fig 2B**); in patients with HLA-DR heterozygosity compared with those with HLA-DR homozygosity (**Fig 2C**); individuals under 18 years of age (**Fig 2D**); patients with positive anti-Hbc antibodies compared to those with negative anti-Hbc antibodies (**Fig 2E**).

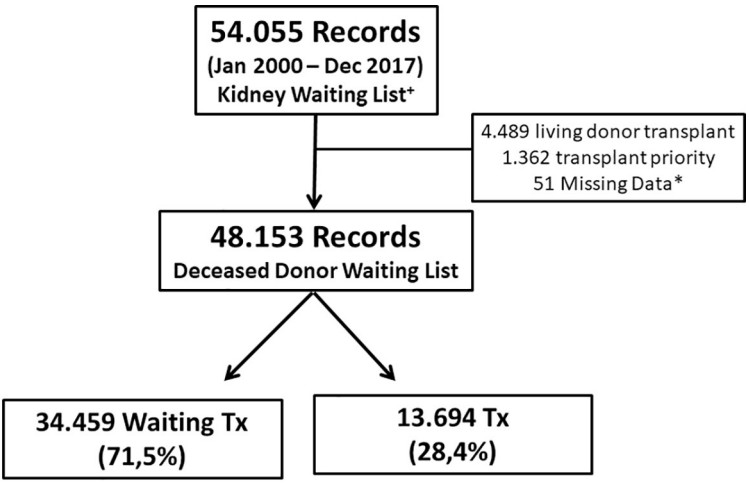

**Fig 1. Patient flowchart.**

## Cox regression model

The Cox model showed that age, subregion, cPRA, and frequency of HLA-DR, -B and -A were associated with changes in likelihood of transplantation (**Table 1**).

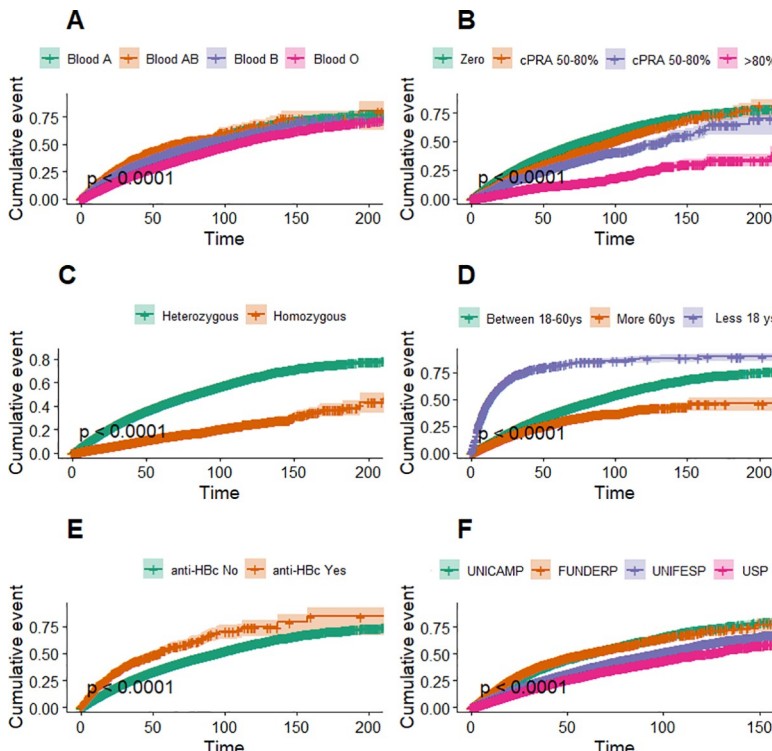

**Fig 2.** Kaplan-Meier plot showing the Likelihood of transplantation in different groups: A: blood groups; B: cPRA (calculated panel class I) strata, C: HLA-DR homozygossity heterozygosity, D: age strata, E: presence of anti-Hbc (Hepatitis B surface antibody); F: between subregions.

**Table 1. Hazard ratios for predictors of deceased donor transplants in univariate and multivariate Cox regression models.**

| | Univariate Cox | | | Multivariate Cox | | |
|---|---|---|---|---|---|---|
| *Predictors* | *Estimates* | *CI* | *p* | *Estimates* | *CI* | *P* |
| Age between 18 and 60 yrs. (reference) | | | | | | |
| Age more than 60 yrs. | 0.68 | 0.65 – 0.72 | <**0.001** | 0.69 | 0.65 – 0.73 | <**0.001** |
| Age less than 18 yrs. | 5.32 | 4.92 – 5.74 | <**0.001** | 5.29 | 4.90 – 5.71 | <**0.001** |
| Male Sex | 0.99 | 0.95 – 1.04 | 0.761 | | | |
| Time on dialysis [log] | 1.06 | 1.04 – 1.09 | <**0.001** | 1.06 | 1.04 – 1.09 | <**0.001** |
| Diabetes | 0.97 | 0.92 – 1.02 | 0.228 | | | |
| Blood group A (reference) | | | | | | |
| Blood group AB | 1.24 | 1.12 – 1.37 | <**0.001** | 1.24 | 1.12 – 1.36 | <**0.001** |
| Blood group B | 0.97 | 0.91 – 1.03 | 0.348 | 0.97 | 0.91 – 1.03 | 0.353 |
| Blood group O | 0.66 | 0.64 – 0.69 | <**0.001** | 0.66 | 0.64 – 0.69 | <**0.001** |
| Prior to Transplant | 0.92 | 0.86 – 0.99 | **0.022** | 0.92 | 0.86 – 0.99 | **0.019** |
| Subregion FUNDERP (reference) | | | | | | |
| Subregion UNICAMP | 1.02 | 0.94 – 1.10 | 0.613 | 1.02 | 0.94 – 1.10 | 0.638 |
| Subregion UNIFESP | 0.65 | 0.62 – 0.69 | <**0.001** | 0.65 | 0.62 – 0.69 | <**0.001** |
| Subregion HCFMUSP | 0.49 | 0.46 – 0.52 | <**0.001** | 0.49 | 0.46 – 0.52 | <**0.001** |
| cPRA zero (reference) | | | | | | |
| cPRA between 0 and 50% | 0.73 | 0.69 – 0.77 | <**0.001** | 0.73 | 0.69 – 0.77 | <**0.001** |
| cPRA between 50 and 80% | 0.57 | 0.51 – 0.63 | <**0.001** | 0.57 | 0.51 – 0.63 | <**0.001** |
| cPRA > 80% | 0.23 | 0.20 – 0.26 | <**0.001** | 0.23 | 0.20 – 0.26 | <**0.001** |
| Anti-HBc | 2.15 | 1.89 – 2.45 | <**0.001** | 2.15 | 1.88 – 2.45 | <**0.001** |
| HLA-DR frequency | 1.08 | 1.06 – 1.09 | <**0.001** | 1.08 | 1.06 – 1.09 | <**0.001** |
| HLA-B frequency | 1.10 | 1.09 – 1.12 | <**0.001** | 1.10 | 1.08 – 1.12 | <**0.001** |
| HLA-A frequency | 1.04 | 1.03 – 1.05 | <**0.001** | 1.04 | 1.03 – 1.05 | <**0.001** |
| HLA-DR homozygosity | 0.36 | 0.32 – 0.40 | <**0.001** | 0.36 | 0.32 – 0.40 | <**0.001** |
| HLA-B homozygosity | 0.79 | 0.71 – 0.87 | <**0.001** | 0.78 | 0.71 – 0.86 | <**0.001** |
| HLA-A homozygosity | 0.96 | 0.89 – 1.03 | 0.241 | | | |
| Observations | 35117 | | | 35117 | | |
| $R^2$ Nagelkerke | 0.403 | | | 0.403 | | |

cPRA: calculated panel class I; Anti-HBc: Hepatitis B surface antibody; HLA: Human leukocyte antigen.

**Model validation.** The final multivariate Cox model was trained in 75% of the data (36,115) and tested in 25% of the data (12,038), with a c-index of 0.70 (c-Index) (**Fig 3**).

## Sensitivity analysis

For the analysis of sensitivity, multivariate Cox models were fitted using stepwise selection for each allocation subregion. Considering three data subsets: FUNDERP + UNICAMP, UNIFESP, and USP. FUNDERP and UNICAMP data were combined into a single subset owing to the low number of cases of these subregions. Despite the small coefficient variations observed among subsets, no major differences in the final coefficients were found compared to the full model (**S2 Table**). C-indexes in the subsets FUNDERP+UNICAMP, USP, and UNIFESP (0.67, 0.69 and 0.68., respectively) were slightly lower than that in the full model.

## Competitive risk analysis

A regression model was fitted using competitive risk as described by Fine & Gray[15]. considering the four different possible outcomes (staying on waiting list, transplant, removal from

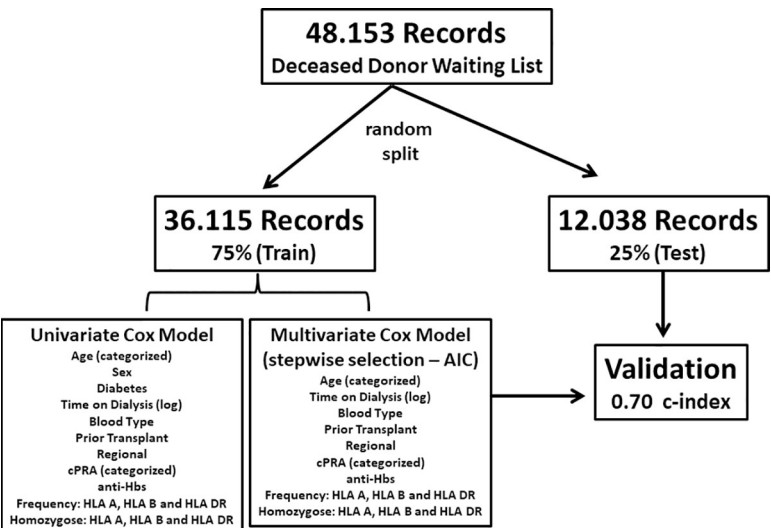

**Fig 3. Workflow of the prediction model on training data and its validation on the testing set.**

waiting list, or death). Thus, the patients removed from the waiting list could be re-enrolled and considered as new cases. Notably, the regression coefficients of the competitive risk model were very close to the Cox regression values (**S3 Table**).

## Transplant waiting list calculator

A calculator of kidney transplant waiting time using Cox model hazard ratios can be accessed at: https://gustavomodelli.shinyapps.io/time_list_in_tx/

## Discussion

Using the database of the largest state in number of transplants in Brazil, which covers a period of 17 years, we developed and validated a model for predicting waiting time to kidney transplantation with a good internal validation concordance (c-index = 0.70). Approximately 1/3 of the patients were transplanted during the study period and the likelihood of transplantation was greater in the first 50 months. The median transplant time was 26 months, about half of the 48 months reported in the American allocation system [20].

The methodology described here considered major factors that could affect waiting time for a deceased donor kidney transplant. As expected, age >18 years was associated with a greater likelihood of transplant, while age>60 years was associated with a lower chance of transplant, possibly due to the higher frequency of deteriorating clinical conditions seen in this age group [21]. Transplant likelihood was higher in the subregional centers where the number of candidates listed was lower. Expectedly, transplantation likelihood increased with HLA frequency, especially HLA-DR, as observed in the American allocation system, which also uses HLA scores [22]. Another significant factor that raised the probability of transplantation was the presence of positive serology for anti-HBc, which can be explained by the fact that our national allocation system offers organs from positive anti-Hbc donors to recipients who are either positive anti-Hbc or vaccinated (anti-Hbs positive) [13].

A lower likelihood of transplantation was seen in Blood group O as reported by others [23], where the export of blood group O donor kidneys to patients with other blood groups or zero mismatches leads to longer waiting times, to a higher death rate and to accumulation of blood

group [23]. A high cPRA (> 80%) was associated with a much lower probability of transplantation (77%) consistently with information provided by the American [22] and French allocation systems [24]. The possibility of transplantation was also decreased among sensitized patients who, despite having higher allocation scores, often showed a positive crossmatch, which explains their reduced chance.

## Prediction model

Prognostic scores currently implemented in transplant medicine mostly predict graft loss, renal function, and the likelihood of transplantation [25–27].

Hart et al. [8] also described a calculator for the likelihood of outcomes for kidney transplant candidates, which includes waiting time after listing for a deceased donor kidney transplant. C statistics for their models was 0.64. In this study, we fit a predictive model for the kidney transplant waiting list using clinical variables available at the time of registration on the waiting list. The model was validated in 25% of the data (internal validation), with a c-index of 0.70. Predictive models are considered useful when the C statistic is greater than 0.70, and strong when the C statistic exceeds 0.80. The iBox (integrative box risk prediction score—iBox), a risk prediction score combining demographic, functional, histological, and immunological factors, showed a c-index greater than 0.80 [28]. A similar study in the American population using the SRTR ("Scientific Registry of Transplant Recipients") database found a c-index ranging from 0.64 to 0.73 [8]. In other fields, such as oncology, prediction results generally range between 0.60 and 0.70 [29]. These results suggest that our estimates are within useful ranges and that the statistical model used here is more accurate and performs better than the method of reporting median time to transplant [8, 21].

## Limitations

This study is limited by the data source used. However, it is worth of note that the São Paulo State Organ Allocation System accounts for 50% of the kidney transplant activity in Brazil.

Cox regressions models could potentially overestimate waiting time because "deaths on the waiting-list" and "removal from the waiting list" are censored or not included in the calculation using this approach. Nonetheless, as the competitive risk model coefficients were very close to Cox regression values, the latter were used. Furthermore, competitive risk models are more difficult to be interpreted by non-specialists [24].

## Practical applications

Our model provides estimates on waiting time for a deceased kidney transplant using hazard ratio. However, it can also calculate the probability of kidney transplant at a specific time point based on Cox survival time (See **S4 Table** for 7 simulated cases). Our online calculator for waiting time to kidney transplant can also be made available on mobile phone applications or incorporated into the transplant database itself. A better estimate of waiting time can improve counselling to kidney transplant candidates, help recruitment for trials, and even lead to changes in allocation strategies.

## Conclusion

Our calculator of kidney transplant waiting time shows good predictive performance and provides information that may be valuable in assisting candidates and their providers in making informed decisions. As the number of patients on the waiting list grows, predicting the time

frame for waiting to a kidney transplant can significantly improve the use of economic resources, as well as the management of patient care before transplant.

## Supporting information

**S1 Table. Characteristics of the patients on the kidney transplant waiting list between January 1, 2000, and December 31, 2017.**
(DOCX)

**S2 Table. Sensitivity analysis of multivariate Cox model for each subregion (FUNDERP + UNICAMP), USP, and UNIFESP.**
(DOCX)

**S3 Table. Competitive risk regression model for transplant considering the removal of the list or death as competitive events and the transplant as the outcome.**
(DOCX)

**S4 Table. Estimation of the probability of transplantation in simulated cases.**
(DOCX)

## Author Contributions

**Conceptualization:** Juliana Feiman Sapiertein Silva.

**Data curation:** Juliana Feiman Sapiertein Silva, Hong Si Nga.

**Formal analysis:** Hong Si Nga, Luis Gustavo Modelli de Andrade.

**Methodology:** Gustavo Fernandes Ferreira, Marcelo Perosa.

**Supervision:** Luis Gustavo Modelli de Andrade.

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
