## [Decision Letter · Decision Letter 0]

8 Apr 2021

PONE-D-21-08778

A Machine Learning Prediction Model for waiting time to kidney transplant

PLOS ONE

Dear Dr. Modelli de Andrade,

Thank you for submitting your manuscript to PLOS ONE. After careful consideration, we feel that it has merit but does not fully meet PLOS ONE’s publication criteria as it currently stands. Therefore, we invite you to submit a revised version of the manuscript that addresses the points raised during the review process.

The authors need to respond to the Reviewers' comments:

Reviewer # 1:

The manuscript presents interesting data on use of machine learning to predict time in waiting list. The authors have performed a model using several demographic data and test in a cohort of patients with success. They created a online live system to be tested. My minor concerns are:

- The intervals of age and cPRA seem pretty broad. Do they tested for more categories in-betweens?

- What about the influence of comorbities in this model: cardiovascular, bone, so on?

- Can this model be tested outside Sao Paulo State and has the same fitness?

Reviewer # 2:

The manuscript by de Andrade et al. is a well designed computational science project aimed at learning to predict the waiting time for deceased donor kidney transplant waiting list patients. Apparently, the output of their statistical models was the probability (hazard ratio) of the transplant, rather than the actual waiting time. Statistical methods predicting actual survival time exist or are possible in theory, and it would be nice to mention in introduction or discussion. There are a few smaller details that can be polished in the manuscript. When saying that patients with missing data were excluded, please indicate: missing data in which (or maybe all?) variables. Also, please mention how did you use allelefrequencies.net database for calculating HLA antigen frequency. In particular, when alleles from the database were mapped to antigens, was care taken to account for broads and splits? Usually people refer to HLA dictionaries (such as the one here: https://pubmed.ncbi.nlm.nih.gov/19140825/) to make sure conversion from allelic to antigenic HLA types was correct.

We look forward to receiving your revised manuscript.

Kind regards,

Stanislaw Stepkowski

Academic Editor

PLOS ONE

Journal Requirements:

In your Methods section, please ensure that sufficient information to make the study reproducible are provided (for example, by describing the models and equations  used, and describing parameters and assumptions applied).

We note that you have indicated that data from this study are available upon request. PLOS only allows data to be available upon request if there are legal or ethical restrictions on sharing data publicly. For information on unacceptable data access restrictions, please see http://journals.plos.org/plosone/s/data-availability#loc-unacceptable-data-access-restrictions.

3a) If there are ethical or legal restrictions on sharing a de-identified data set, please explain them in detail (e.g., data contain potentially identifying or sensitive patient information) and who has imposed them (e.g., an ethics committee). Please also provide contact information for a data access committee, ethics committee, or other institutional body to which data requests may be sent.

3b) If there are no restrictions, please upload the minimal anonymized data set necessary to replicate your study findings as either Supporting Information files or to a stable, public repository and provide us with the relevant URLs, DOIs, or accession numbers. Please see http://www.bmj.com/content/340/bmj.c181.long for guidelines on how to de-identify and prepare clinical data for publication. For a list of acceptable repositories, please see http://journals.plos.org/plosone/s/data-availability#loc-recommended-repositories.

Additional Editor Comments :

The authors need to respond to the Reviewers' comments:

Reviewer # 1:

The manuscript presents interesting data on use of machine learning to predict time in waiting list. The authors have performed a model using several demographic data and test in a cohort of patients with success. They created a online live system to be tested. My minor concerns are:

- The intervals of age and cPRA seem pretty broad. Do they tested for more categories in-betweens?

- What about the influence of comorbities in this model: cardiovascular, bone, so on?

- Can this model be tested outside Sao Paulo State and has the same fitness?

Reviewer # 2:

The manuscript by de Andrade et al. is a well designed computational science project aimed at learning to predict the waiting time for deceased donor kidney transplant waiting list patients. Apparently, the output of their statistical models was the probability (hazard ratio) of the transplant, rather than the actual waiting time. Statistical methods predicting actual survival time exist or are possible in theory, and it would be nice to mention in introduction or discussion. There are a few smaller details that can be polished in the manuscript. When saying that patients with missing data were excluded, please indicate: missing data in which (or maybe all?) variables. Also, please mention how did you use allelefrequencies.net database for calculating HLA antigen frequency. In particular, when alleles from the database were mapped to antigens, was care taken to account for broads and splits? Usually people refer to HLA dictionaries (such as the one here: https://pubmed.ncbi.nlm.nih.gov/19140825/) to make sure conversion from allelic to antigenic HLA types was correct.

Reviewers' comments:

Reviewer's Responses to Questions

**Comments to the Author**

1. Is the manuscript technically sound, and do the data support the conclusions?

Reviewer #1: Yes

Reviewer #2: Yes

2. Has the statistical analysis been performed appropriately and rigorously? 

Reviewer #1: I Don't Know

Reviewer #2: Yes

3. Have the authors made all data underlying the findings in their manuscript fully available?

Reviewer #1: Yes

Reviewer #2: Yes

4. Is the manuscript presented in an intelligible fashion and written in standard English?

Reviewer #1: Yes

Reviewer #2: Yes

5. Review Comments to the Author

Reviewer #1: The manuscript presents interesting data on use of machine learning to predict time in waiting list. The authors have performed a model using several demographic data and test in a cohort of patients with success. They created a online live system to be tested. My minor concerns are:

- The intervals of age and cPRA seem pretty broad. Do they tested for more categories in-betweens?

- What about the influence of comorbities in this model: cardiovascular, bone, so on?

- Can this model be tested outside Sao Paulo State and has the same fitness?

Reviewer #2: The manuscript by de Andrade et al. is a well designed computational science project aimed at learning to predict the waiting time for deceased donor kidney transplant waiting list patients. Apparently, the output of their statistical models was the probability (hazard ratio) of the transplant, rather than the actual waiting time. Statistical methods predicting actual survival time exist or are possible in theory, and it would be nice to mention in introduction or discussion. There are a few smaller details that can be polished in the manuscript. When saying that patients with missing data were excluded, please indicate: missing data in which (or maybe all?) variables. Also, please mention how did you use allelefrequencies.net database for calculating HLA antigen frequency. In particular, when alleles from the database were mapped to antigens, was care taken to account for broads and splits? Usually people refer to HLA dictionaries (such as the one here: https://pubmed.ncbi.nlm.nih.gov/19140825/) to make sure conversion from allelic to antigenic HLA types was correct.

6. PLOS authors have the option to publish the peer review history of their article (what does this mean?). If published, this will include your full peer review and any attached files.

Reviewer #1: No

Reviewer #2: **Yes: **Dulat Bekbolsynov

---

## [Author Response · Author response to Decision Letter 0]

21 Apr 2021

Reviewer # 1:

The manuscript presents interesting data on use of machine learning to predict time in waiting list. The authors have performed a model using several demographic data and test in a cohort of patients with success. They created a online live system to be tested. 

Thank you very much for your comments

My minor concerns are:

- The intervals of age and cPRA seem pretty broad. Do they tested for more categories in-betweens?

We fitted the cox model using cPRA and age at different cut-offs, which resulted in values of similar accuracy (c-index). Then we chose simple three-class values for age (< 18, between 18 and 60 and above 60 ys) and 4 class values for cPRA (zero, between o and 50%, between 50 and 80%, and above 80%). For age, these categories were chosen using the WHO cut-offs (children/older). 

- What about the influence of comorbities in this model: cardiovascular, bone, so on?

We fitted the model with all variables available on the waiting list registration records. These variables are age, race, underlying diseases, cPRA, HLA, serology, and time on dialysis. Unfortunately, we did not have any comorbidities. The only comorbidity predictor available was diabetes. Diabetes was not an independent predictor of waiting time in the final model. A complete list of predictors is avaiable in the attachment table 01.

- Can this model be tested outside Sao Paulo State and has the same fitness?

Probably so, provided some small adjustments are made in the model

Reviewer # 2:

The manuscript by de Andrade et al. is a well designed computational science project aimed at learning to predict the waiting time for deceased donor kidney transplant waiting list patients. Apparently, the output of their statistical models was the probability (hazard ratio) of the transplant, rather than the actual waiting time. Statistical methods predicting actual survival time exist or are possible in theory, and it would be nice to mention in introduction or discussion.

Thank you very much for your comments

Yes, our model provides estimates on waiting time for a deceased kidney transplant using hazard ratio. However, it can also calculate the probability of a kidney transplant at a specific time point based on Cox survival time. We provided an attachment which shows the prediction of the probability of transplant at different time points in simulated cases (attachment 04). 

We used the equations to calculate probabilities:

Hazard Ratio: The hazard ratio is the ratio of these two expected hazards: h0(t)exp (b1a)/ h0(t)exp (b1b) = exp(b1(a-b)) which does not depend on time, t. Thus the hazard is proportional over time

Probability of transplant at any time: proportion of units that survive beyond a specified time

Emil Hvitfeldt provided an excellent example using the Cox model to obtain the prediction of time and survival (https://github.com/tidymodels/censored)

 There are a few smaller details that can be polished in the manuscript. When saying that patients with missing data were excluded, please indicate: missing data in which (or maybe all?) variables. 

Data were excluded when subregional information was missing (n=51). We provided this information in the text. 

Also, please mention how did you use allelefrequencies.net database for calculating HLA antigen frequency. In particular, when alleles from the database were mapped to antigens, was care taken to account for broads and splits? Usually people refer to HLA dictionaries (such as the one here: https://pubmed.ncbi.nlm.nih.gov/19140825/) to make sure conversion from allelic to antigenic HLA types was correct.

We clarifed HLA frequency calculation methods in the manuscript.

 HLA frequency in the population of São Paulo state was obtained from the Allele Frequencies net database. To standardize HLA antigen assignments an HLA Dictionary [01] was used, and the serological equivalents were listed as expert assigned types.

01. Holdsworth R, Hurley CK, Marsh SG, Lau M, Noreen HJ, Kempenich JH, et al. The HLA dictionary 2008: a summary of HLA-A, -B, -C, -DRB1/3/4/5, and -DQB1 alleles and their association with serologically defined HLA-A, -B, -C, -DR, and -DQ antigens. Tissue Antigens. 2009;73(2):95-170. doi: 10.1111/j.1399-0039.2008.01183.x. PMID: 19140825

---

## [Editor Report · Decision Letter 1]

10 May 2021

A Machine Learning Prediction Model for waiting time to kidney transplant

PONE-D-21-08778R1

Dear Dr. Modelli de Andrade,

We’re pleased to inform you that your manuscript has been judged scientifically suitable for publication and will be formally accepted for publication once it meets all outstanding technical requirements.

Kind regards,

Stanislaw Stepkowski

Academic Editor

PLOS ONE
---

## [Editor Report · Acceptance letter]

12 May 2021

PONE-D-21-08778R1 

A Machine Learning Prediction Model for waiting time to kidney transplant 

Dear Dr. de Andrade:

I'm pleased to inform you that your manuscript has been deemed suitable for publication in PLOS ONE. Congratulations! Your manuscript is now with our production department. 

Kind regards, 

on behalf of

Dr. Stanislaw Stepkowski 

Academic Editor

PLOS ONE